# Investigation of the Multi-Scale Deterioration Mechanisms of Anhydrite Rock Exposed to Freeze–Thaw Environment

**DOI:** 10.3390/ma17030726

**Published:** 2024-02-02

**Authors:** Xiaoguang Jin, Chao Hou, Jie He, Daniel Dias

**Affiliations:** 1Laboratory of New Technology for Construction of Cities in Mountain Area of the Ministry of Education, Chongqing University, Chongqing 400045, China; 2State Key Laboratory of Coal Mine Disaster Dynamics and Control, Chongqing University, Chongqing 400045, China; 3School of Civil Engineering, Chongqing University, Chongqing 400045, China; 20201601069@cqu.edu.cn; 4School of Civil Engineering and Architecture, Henan University of Science and Technology, Luoyang 471000, China; chaohou@haust.edu.cn; 5Laboratory 3SR, Grenoble Alpes University, CNRS UMR 5521, 38000 Grenoble, France; daniel.dias@univ-grenoble-alpes.fr

**Keywords:** freeze–thaw cycles, physical and mechanical properties, multi-scale, deterioration mechanisms, anhydrite rock

## Abstract

The deterioration of anhydrite rock exposed to a freeze–thaw environment is a complex process. Therefore, this paper systematically investigated the physical and mechanical evolutions of freeze–thawed anhydrite rock through a series of multi-scale laboratory tests. Meanwhile, the correlation between pore structure and macroscopic mechanical parameters was discussed, and the deterioration mechanisms of anhydrite rock under freeze–thaw cycles were revealed. The results show that with the increase in freeze–thaw processes, the mechanical strength, elastic modulus, cohesion, proportions of micropores (r ≤ 0.1 μm), and PT-Ipore throat (0–0.1 μm) decrease exponentially. In comparison, the mass variation, proportions of mesopores (0.1 μm < r < 1 μm), macropores (r ≥ 1 μm), and PT-II pore throat (0.1–4 μm) increase exponentially. After 120 cycles, the mean porosity increases by 66.27%, and there is a significant honeycomb and pitted surface phenomenon. Meanwhile, as the freeze–thaw cycles increase, the frost resistance coefficient decreases, while the damage variable increases. The correlation analysis between pore structure and macroscopic mechanical parameters shows that macropores play the most significant role in the mechanical characteristic deterioration of freeze–thawed anhydrite rock. Finally, it is revealed that the water–rock expansion and water dissolution effects play a crucial role in the multi-scale damage of anhydrite rock under the freeze–thaw environment.

## 1. Introduction

Cold regions are widely distributed worldwide, accounting for approximately 75% of China’s land area [1]. Rocks are formed by mineral grains, pores, and microcracks [2,3], and the characteristics of rocks are greatly influenced by the environment they sustain [4,5,6]. With the development of transportation, infrastructure, and energy facilities, numerous rock engineering projects, such as tunnels, highways, slopes, and open pit mines, are planned or constructed in cold regions [7,8,9]. The temperature in the cold areas fluctuates around 0 °C with the change in seasons and the transition of day and night, leading to periodic freezing and thawing of water in the rock, and the physical and mechanical properties of rocks can be irreversibly damaged under freeze–thaw cycles [10,11,12,13]. Thus, rock engineering located in cold areas is exposed to a freeze–thaw environment, which is a severe threat to the safe construction and healthy operation of the engineering project [14,15]. As a result, the multi-scale physical and mechanical characteristics of rocks, such as strength, durability, and porosity, have been investigated by researchers for decades [13,16,17].

The effects of freeze–thaw cycles on the mechanical characteristics of rocks were investigated through a series of laboratory experiments [18,19,20,21]. For example, Mousavi et al. [22] measured the physical and mechanical properties of calc-schist rock samples from an open-pit mine under frozen and thawed processes. They indicated that the main mechanical properties of calc-schist rock, such as cohesion, internal friction, and triaxial compression strength, decrease with freeze–thaw cycles. Zhang et al. [23] tested the mechanical properties of sandstones that experienced different freeze–thaw cycles with four kinds of confining pressures and analyzed the whole process characteristics of the deformation and failure of sandstone. Wang et al. [24] selected fine sandstone and coarse sandstone as representatives of hard rocks and studied physical and triaxial compression mechanical properties under the freeze–thaw environment. Fang et al. [25] investigated the coupling effects of chemical corrosion and freeze–thaw cycles on the mechanical characteristics of yellow sandstone. They concluded that both chemical corrosion and freeze–thaw processes adversely impact the rock’s mechanical characteristics. Mu et al. [26] conducted shear tests on three typical types of jointed rocks under the freeze–thaw environment and examined the degradation characteristics through the evolution of shear parameters. Luo et al. [27] analyzed the dynamic mechanical properties of rock under dynamic loading and freeze–thaw action and obtained the static and dynamic mechanical parameters of sandstone. Martinez-Martinez et al. [28] conducted a long-term test to investigate the resistances of six different limestone types, and the evolution of their properties under the freeze–thaw weathering process was obtained.

The microstructure of rocks directly impacts the mechanical characteristics. Thus, the micropores and cracks induced by freeze–thaw cycles can also be detected through nondestructive tests such as Nuclear Magnetic Resonance (NMR) and Scanning Electron Microscope (SEM) [29,30]. Liu et al. [31] used the NMR technique to study the freeze–thaw damage degradation of sandstone with initial damage, and the porosity, T_2_ spectrum distribution, and the T_2_ spectral area were obtained. Tian et al. [32] used NMR and SEM techniques to monitor the micropore evolution of three soils under repeated freeze–thaw cycles. Martínez-Martínez et al. [28] conducted SEM tests to study the microstructure evolution of polished rocks after freeze–thaw cycles. Ke et al. [33] conducted NMR tests and impact loading experiments on sandstone under different freeze–thaw cycles. They indicated that with the increase in freeze–thaw processes, the pore expands, and the pore size tends to be uniform. Park et al. [34] conducted artificial weathering tests in the laboratory and SEM images were obtained to investigate the microstructure evolution of diorite, basalt, and tuff specimens with freeze–thaw cycles.

The previous articles from the literature presented the physical and mechanical deterioration of various rocks exposed to freeze–thaw cycles. However, previous research has mainly concentrated on sandstone, granite, marble, limestone, quartz, shale, amphibolite, diorite, basalt, tuff, etc. [16,19,35,36]. Few studies have involved the multi-scale deterioration mechanisms of anhydrite rock under a freeze–thaw environment. It has been reported that anhydrite rock will swell and dissolve when encountering water, causing a decrease in its bearing capacity and durability [37]. Consequently, it is determined that the freeze–thawed damage mechanisms of anhydrite rock are different from other rocks. Thus, in this paper, a series of multi-scale laboratory tests were conducted systematically to reveal the deterioration mechanisms of anhydrite rock exposed to freeze–thaw cycles. The multi-scale physical and mechanical evolution characteristics were obtained, and the relationship between the pore structure and mechanical strength was established. Finally, the deterioration mechanisms of anhydrite rock exposed to the freeze–thaw environment were discussed.

## 2. Materials and Methods

### 2.1. Sample Preparation

The anhydrite rock studied in this paper was taken from Donghuishe Town, Pingshan County, Hebei Province, where the annual extreme low temperature is −16.4 °C. The sample appears light gray under natural conditions, with an apparent density of 2.93 g/cm^3^ and a porosity of 0.69%. X-ray diffraction (XRD) results showed that the main mineral components of anhydrite rock are anhydrite (81%), dihydrate gypsum (9%), dolomite (7%), and 3% other minerals (Figure 1). In this paper, three different specifications of samples were used, namely: cylindrical samples with a size of Φ 50 × H 100 mm (used for mass variation tests, uniaxial and triaxial compression tests, and NMR analysis), cylindrical samples with a size of Φ 50 × H 25 mm (used for Brazilian splitting tests), and rectangular samples with a size of L10 × W10 × H5 mm (used for SEM tests). It should be noted that the rectangular samples used for the SEM tests were taken from the interior of the rock block to avoid the impact of the natural surface weathering effect.

### 2.2. Test Scheme and Main Instruments

The test scheme for freeze–thaw was to divide the samples into five groups and conduct 0, 30, 60, 90, and 120 freeze–thaw cycles on each group. Based on the local annual extreme temperature and according to the Chinese code DL/T 5368-2007 (2007) [38], the freeze–thaw temperature was set to −20~+20 °C. One freeze–thaw cycle process is as follows: the water-saturated rock is frozen for 4 h in a freezing environment at −20 °C and then thawed for 4 h in distilled water at 20 °C. After the sample reached the set number of freeze–thaw cycles, it was taken to the corresponding laboratory at room temperature around 20 °C, for a series of tests to obtain the muti-scale physical and mechanical characteristics.

The freeze–thaw cycle test adopted a high- and low-temperature alternating damp heat test system (Figure 2a), with a temperature range of −40~+100 °C. The uniaxial compression tests were performed through the WDAJ-600 microcomputer-controlled electro-hydraulic servo rock shear rheological test machine (Figure 2b), with a maximum axial test force of 600 kN. The triaxial compression tests were carried out through the ROCK600-50HTPLUS multifunctional rock full-stress triaxial test system (Figure 2c), with a maximum axial force of 1000 kN. The Brazilian splitting tests were implemented through the AG-250 kN IS precision electronic universal material test machine (Figure 2d), with a maximum axial force of 250 kN. The NMR test adopted the non-destructive NMR system for rock core imaging and analysis system (Figure 2e), with a magnetic field strength of 0.3 ± 0.05 T, and the pore size, ranging from nano to micrometer scales, was measured. The SEM tests were carried out through a field emission transmission electron microscope (Figure 2f) with a maximum acceleration voltage of 200 kV.

### 2.3. Macro-Scale Tests

#### 2.3.1. Mass Variation Tests

Four cylindrical samples numbered M_1_, M_2_, M_3_, and M_4_ were used for mass variation tests. After every 30 freeze–thaw cycles, the samples were taken out of the water, the surface moisture was wiped away, and the sample’s mass was measured via a scale with an accuracy of 0.01 g. The freeze–thawed mass variation can be calculated according to Equation (1):(1)$$\mathrm{mass}\;\mathrm{variation}\left(\%\right)=\frac{m_{0}-m_{N}}{m_{0}}\times 100\%$$
where m_0_ is the sample’s mass non-exposed to the freeze–thaw environment, and m*_N_* is the sample’s mass after *N* freeze–thaw cycles.

#### 2.3.2. Mechanical Resistance Tests

Fifteen cylindrical samples with a size of Φ 50 × H 100 mm were used for uniaxial compression tests. These samples were divided into five groups, with three in each group. Meanwhile, the displacement loading method was adopted in the uniaxial compression tests with a loading rate of 0.1 mm/min. Twenty cylindrical samples with a size of Φ 50 × H 100 mm were used for triaxial compression tests. Freeze–thaw processes usually occur in the shallow rock strata, where the confining pressure is not very high. Thus, based on the performance of the testing equipment and the mechanical properties of anhydrite rock, two groups of confining pressures (3 MPa, 5 MPa) were set for the triaxial compression tests. Meanwhile, the samples used for the triaxial compression tests were divided into five groups, with four samples in each group (two for the confining pressure of 3 MPa and two for the confining pressure of 5 MPa). In the triaxial compression tests, the stress loading method was first used to load the confining pressure with a loading rate of 0.5 MPa/min. Then, the displacement loading method was used to load the axial stress with a loading rate of 0.1 mm/min. Fifteen cylindrical samples with a size of Φ 50 × H 25 mm were used for the Brazilian splitting tests. They were also divided into five groups, with three in each group. The Brazilian splitting tests were carried out using an arc-shaped mold, and the loading method was displacement loading with a rate of 0.1 mm/min. All the mechanical resistance tests were loaded until the samples reached failure. Then, the mechanical parameters of samples under various freeze–thaw cycles were obtained.

### 2.4. Micro-Scale Tests

Three cylindrical samples with a size of Φ 50 × H 100 mm were used for the NMR tests and were numbered N_1_, N_2_, and N_3_. Before the NMR tests, the samples were vacuumed and pressurized for water saturation. The vacuuming time was set as 30 min, the water saturation pressure was 5 MPa, and the water saturation time was 12 h. It should be noted that a calibration test is required to establish the relationship between the nuclear magnetic unit volume signal and the porosity of the sample at the first NMR test. Three rectangular samples of L10 × W10 × H5 mm were used for the SEM tests and numbered S_1_, S_2_, and S_3_. Before conducting the SEM tests, the samples were placed in an oven for 12 h at a drying temperature of 50 °C. Due to the lack of conductivity of the rock, the surface of the dried sample was sprayed with gold for 60 s.

## 3. Experimental Results and Analysis

### 3.1. Macroscopic Damage Evolution

#### 3.1.1. Mass Variation

The mass variation is an important parameter for evaluating freeze–thaw damage. The freeze–thawed mass variation of four samples calculated by Equation (1) is shown in Figure 3. It can be seen from Figure 3a that the mass variation increases with the number of freeze–thaw cycles, indicating that the samples’ mass has reduced. After 30 freeze–thaw cycles, the mass variation of M_1_, M_2_, M_3_, and M_4_ are 0.28%, 0.16%, 0.23%, and 0.33%, respectively. After 120 freeze–thaw cycles, the mass variation of M_1_, M_2_, M_3_, and M_4_ are 1.14%, 0.68%, 0.47%, and 0.86%, respectively. During the freeze–thaw tests, white powdery or flaky rock solutes can be observed in water. Therefore, we can infer that the decrease in mass of anhydrite rock under freeze–thaw cycles is mainly due to the dissolution and peeling of rock minerals in water. Meanwhile, from Figure 3b, it can be seen that the mass variation mean value exponential grows with the number of freeze–thaw cycles.

#### 3.1.2. Deterioration of Mechanical Characteristics

The results of the uniaxial compression tests of anhydrite rock are shown in Table 1. The evolution of uniaxial compression strength (UCS) and elastic modulus of anhydrite rock with the number of freeze–thaw cycles are shown in Figure 4. It can be seen that as the number of freeze–thaw cycles increases, the UCS and elastic modulus of anhydrite rock decrease exponentially. After 30, 60, 90, and 120 cycles, the mean UCSs decrease by 13.88%, 27.23%, 41.18%, and 46.54%, respectively, and the mean elastic moduli decrease by 25.14%, 38.65%, 52.66%, and 60.16%, respectively.

The results of the triaxial compression tests of anhydrite rock under different confining pressures are shown in Table 2 and Table 3. The evolution of triaxial compression strength (TCS) and elastic modulus of anhydrite rock with the number of freeze–thaw cycles are shown in Figure 5 and Figure 6. As the number of freeze–thaw cycles increases, the TCS and elastic modulus of anhydrite rock under different confining pressures decrease exponentially. When the confining pressure is 3 MPa, after 30, 60, 90, and 120 cycles, the mean TCSs decrease by 15.92%, 25.39%, 30.84%, and 35.72%, respectively, and the mean elastic moduli decrease by 17.36%, 40.23%, 48.64%, and 52.16%, respectively. When the confining pressure is 5 MPa, after 30, 60, 90, and 120 cycles, the mean TCSs decrease by 10.61%, 17.31%, 23.39%, and 28.24%, respectively, and the mean elastic moduli decrease by 17.38%, 33.21%, 42.39%, and 43.89%, respectively.

Based on the uniaxial and triaxial compression results, the Mohr–Coulomb criterion was used to calculate anhydrite rock’s cohesion and internal friction angle under different freeze–thaw cycles, as shown in Table 4. It can be seen from Table 4 that the cohesion decreases with the freeze–thaw cycles. After 30, 60, 90, and 120 cycles, the cohesion decreases by 12.99%, 33.61%, 47.31%, and 52.48%, respectively. The internal friction angle has no noticeable change, and its evolution amplitude is within 4.04°. The internal frictions of granite [39] and sandstone [24] under freeze–thaw cycles also exhibit similar evolution patterns. The above phenomenon is because the internal friction angle mainly reflects the friction characteristics between loose granular rock and soil. However, anhydrite rock, granite, and sandstone are hard rocks, and the freeze–thaw treatment has little effect on their internal friction angle. Meanwhile, Figure 7 shows the variation in cohesion with the number of freeze–thaw cycles. It can be seen that as the number of freeze–thaw cycles increase, the cohesion decreases exponentially.

The test results of Brazilian splitting are shown in Table 5. Meanwhile, Figure 8 shows the evolution of the tension strength and elastic modulus of anhydrite rock with the number of freeze–thaw cycles. It can be seen that the tension strength and elastic modulus exhibit an exponentially decreasing relationship with the increase in freeze–thaw cycles. After 30, 60, 90, and 120 cycles, the mean tension strengths decrease by 21.3%, 35.49%, 45.33%, and 52.8%, respectively, and the mean tension elastic moduli decrease by 29.6%, 35.02%, 58.48%, and 73.65%, respectively.

#### 3.1.3. Macroscopic Damage Processes

The damage phenomenon of a representative sample during the freeze–thaw treatment is shown in Figure 9. It can be seen from Figure 9 that when the number of freeze–thaw cycles was 70, two small-sized cracks appeared in the upper middle of the sample. When the number of cycles reached 90, the width of the cracks in the upper middle increased significantly, and some small-sized cracks also appeared in the lower part of the sample. As the number of cycles continued to grow, at 105 cycles, the macroscopic cracks in the upper and middle parts of the sample penetrated, causing a loss of load-bearing capacity, and the number of cracks in the lower part also significantly increased. It can be seen that the freeze–thaw cycles can cause irreversible damage to anhydrite rock. Thus, it is crucial to reveal the damage evolution process of freeze–thawed anhydrite rock from a macroscopic perspective.

The frost resistance of rocks can be measured by the frost resistance coefficient, which can be calculated by Equation (2):(2)$$R=\frac{\sigma_{N}}{\sigma_{I}}\times 100\%$$
where R is the frost resistance coefficient, $\sigma_{I}$ is the strength of the untreated samples, and $\sigma_{N}$ is the strength of samples treated with *N* freeze–thaw cycles.

Figure 10 shows the variation in the frost resistance coefficient of anhydrite rock under different freeze–thaw cycles. R_c0_, R_c3_, R_c5_, and R_t_ represent the frost resistance coefficient under uniaxial compression tests, triaxial compression tests ($\sigma_{3}=3\;\mathrm{MPa},\;\sigma_{3}=5\;\mathrm{MPa}$), and Brazilian splitting tests, respectively. It can be seen that the frost resistance coefficient of anhydrite rock decreases with the increase in freeze–thaw cycles. After 120 cycles, R_c0_, R_c3_, R_c5_, and R_t_ decrease from 100% to 53.47%, 64.28%, 71.76%, and 47.21%, respectively. Under the same freeze–thaw cycles, the frost resistance coefficient increases with the increase in confining pressure. The frost resistance coefficient of anhydrite rock under tension stress is the smallest, indicating that the freeze–thawed anhydrite rock is unstable under the tension conditions.

Combining the research results of Lemaitre [40], the macroscopic damage variable of anhydrite rock under freeze–thaw cycles was obtained based on the elastic modulus. The calculation method for macroscopic damage variables is as follows:(3)$$D_{E}=\frac{E_{0}-E_{N}}{E_{0}}$$
where $D_{E}$ is the macroscopic damage variable, $E_{0}$ is the elastic modulus of the untreated samples, and $E_{N}$ is the elastic modulus of samples treated with *N* freeze–thaw cycles.

The relationship between the macroscopic damage variable of anhydrite rock and the number of freeze–thaw cycles is shown in Figure 11. D_E0_, D_E3_, D_E5_, and D_Et_ represent macroscopic damage variables of anhydrite rock under uniaxial compression tests, triaxial compression tests ($\sigma_{3}=3\;\mathrm{MPa},\;\sigma_{3}=5\;\mathrm{MPa}$), and Brazilian splitting tests, respectively. It can be seen that the macroscopic damage variables increase with the number of freeze–thaw cycles. After 120 cycles, D_E0_, D_E3_, D_E5_, and D_Et_ increase to 0.60, 0.54, 0.44, and 0.73, respectively. Under the compression stress conditions, the macroscopic damage variable decreases with the confining pressure. For example, after 120 cycles, D_E0_ is 1.11 times higher than D_E3_ and 1.36 times higher than D_E5_. The reason for the above phenomenon is that confining pressure inhibits the development of freeze–thaw damage in anhydrite rocks. In addition, the tension damage variable is greater than the compression damage variable. For example, after 120 cycles, D_Et_ is 1.22, 1.35, and 1.66 times higher than D_E0_, D_E3_, and D_E5_, respectively. It can be noted that the tension strength of rocks is much lower than the compression strength, and rocks are more prone to failure under tension stress. Thus, the damage development for rocks under tension conditions is faster, and the damage is more significant.

#### 3.1.4. Cracking Modes under Different Stress Conditions

The cracking modes of freeze–thawed anhydrite rock under different stress conditions are shown in Figure 12. It can be seen from Figure 12a that the cracking modes of anhydrite rock under uniaxial compression stress mainly exhibit tension cracks or tension–shear cracks. White powdery minerals and a large amount of rock debris can be observed on the fracture surface, accompanied by a block spalling phenomenon. The tension and shear cracks originate from the end face of the sample and continue to develop and converge under axial force, ultimately forming a crack surface, accompanied by the generation of secondary cracks during the main crack propagation process. The reason for this is that anhydrite rock shows brittle characteristics under uniaxial compression conditions.

It can be seen from Figure 12b that anhydrite rock exhibits three crack modes under triaxial compression conditions: shear crack, tension–shear crack, and tension crack. The cracks are generated at the end face of the sample and are accompanied by a small number of secondary cracks. The failure morphology of the sample is relatively integral compared to that under uniaxial compression.

It can be seen from Figure 12c that the cracking modes of anhydrite rock under tension stress can be divided into three types: linear, arc, and composite types. Linear-type cracks are generated near the upper and lower loading points of the sample and are distributed in the center of the disc. They evenly split the specimen into two parts and are often accompanied by secondary cracks at the ends of linear cracks due to end constraints. Arc-type cracks also originate from the upper and lower loading points of the sample, and the cracks protrude to the right in a circular arc shape, accompanied by secondary cracks at the end of the main cracks. Composite-type cracks are formed by the convergence and connection of multiple cracks, with complex crack morphology and a branching distribution pattern.

Ban [41] pointed out that due to the substantial homogeneity of sandstone, the maximum horizontal tension strain occurs at the center of the specimen, and the failure mode is relatively simple, which is a central crack connecting the upper and lower loading points in an approximate straight line. The anhydrite rock disc sample studied in this article exhibits multiple cracking forms because there are many natural joints in the anhydrite rock. Although the samples are selected in the early stages of the experiment to ensure homogeneity, heterogeneity still exists. When the crystal homogeneity of the anhydrite rock minerals is high, cracks develop along the center of the sample under the tension stress, showing an approximate linear type crack. Suppose the homogeneity of the sample is slightly low when the crack originates from the upper and lower loading points and gradually expands downwards; in that case, it will bypass the higher-strength mineral crystal and develop in other directions, forming an arc or composite-type cracks.

### 3.2. Microscopic Damage Analysis

#### 3.2.1. NMR Test Results

Porosity is one of the most important indicators for evaluating the physical and mechanical properties and pore structure characteristics of rocks. The changes in the porosity of anhydrite rock under freeze–thaw cycles detected by NMR tests are shown in Figure 13. It can be seen that the porosity of the N_1_, N_2_, and N_3_ samples increases exponentially with the number of freeze–thaw cycles and has a good correlation (R^2^ = 0.97). After 30, 60, 90, and 120 cycles, the porosity of the N_1_ sample increases by 21.88%, 42.19%, 60.94%, and 75%, respectively, and the porosity of the N_2_ sample increases by 8.22%, 27.4%, 41.1%, and 57.53%, respectively. It should be noted that due to the failure of the N_3_ sample during 90 to 120 cycles, further NMR testing cannot be conducted. Therefore, the data at 120 cycles of the N_3_ sample are not provided. After 30, 60, and 90 cycles, the porosity of the N_3_ sample increases by 15.71%, 31.43%, and 40%, respectively.

NMR testing usually uses T_2_ relaxation time to reflect the distribution of pore structure, and the T_2_ relaxation time has the advantages of fast monitoring speed and high accuracy. Due to the similarity in pore structure evolutions of different samples, the N_1_ sample was selected as a representative for analysis, and the pore size distribution of the N_1_ sample detected by NMR tests is shown in Figure 14.

Figure 14 shows that the pore size of anhydrite rock under freeze–thaw cycles is mainly distributed between 0.001 and 25 μm and presents a “three peak” distribution, indicating that the pore distribution of anhydrite rock is relatively complex. The peak area between 1 and 10 μm is more significant than the other two peak areas, indicating that the pore distribution of anhydrite rock is uneven. A reasonable pore size classification needs to be proposed to further analyze the pore size evolution for anhydrite rock under freeze–thaw cycles. The pores of anhydrite rocks are divided into three types: micropores (r ≤ 0.1 μm), mesopores (0.1 μm < r < 1 μm), and macropores (r ≥ 1 μm).

The pore volume and proportion can be directly reflected through the curve area in Figure 14, and the curve area evolution for different types of pores of the N_1_ sample is depicted in Figure 15. As shown in Figure 15, with the increase in freeze–thaw cycles, the curve area of micropores in anhydrite rock decreases exponentially. In contrast, the curve area of mesopores and macropores increases exponentially. The correlation coefficient of the fitting curve is greater than 0.72. After 120 cycles, the areas of the micropores decrease by 64.71%, while the areas of the mesopore and macropore increase by 71.28% and 117.32%, respectively. Based on this, it can be deduced that under the freeze–thaw environment, the micropores in anhydrite rock gradually transform into mesopores and macropores, causing a decrease in the volume of micropores and an increase in the volume of mesopores and macropores. In addition, the increase in macropores is much more significant than that of mesopores.

The pore structures of rocks include pores and pore throats, as shown in Figure 16. Pores can store fluid substances such as water, and the pore throats are the connecting channels between pores. The existence of pore throats provides a pathway for fluid migration within rocks. It is believed that the fewer the pore throats, the poorer the connectivity of the crack network and the higher the tortuosity. Thus, the distribution of pore throats has an essential impact on the permeability characteristics and porosity of rock. The evolution of pore throat distribution in anhydrite rock under freeze–thaw cycles is shown in Figure 17. It can be seen that under different freeze–thaw processes, the pore throat diameter distribution of anhydrite rock is within the range of 0–25 μm. The proportion of pore throats in 0–0.1 μm is the highest, and the proportion of pore throats in 16–25 μm is the smallest.

Similarly, the pore throats of anhydrite rock are divided into three types according to their size, namely: PT-I (0–0.1 μm), PT-II (0.1–4 μm), and PT-III (4–25 μm). Figure 18 shows the variation in the proportion of different types of pore throats of the N_1_ samples. It can be seen that there is an excellent exponentially decreasing trend between the PT-I pore throat proportion and the number of freeze–thaw cycles (R^2^ = 0.98). At the same time, there is a perfect exponentially increasing trend between the PT-II pore throat proportion and the number of freeze–thaw cycles (R^2^ = 0.83). After 120 cycles, the proportion of PT-I pore throats decreases by 65.89%, while the proportion of PT-II pore throats increases by 54.54%. The above phenomenon indicates that as the number of freeze–thaw cycles increases, the diameter of some PT-I pore throats gradually increases, transforming into PT-II pore throats. Due to the lack of a clear variation relationship between the PT-III pore throat and the number of freeze–thaw cycles, the PT-III pore throat is not plotted there.

#### 3.2.2. SEM Test Results

Figure 19 shows the SEM images of the morphological evolution of the anhydrite samples under a freeze–thaw action. It should be noted that it is challenging to achieve positioning observation during the freeze–thaw tests because the positioning observation marks easily fall off and disappear under the freeze–thaw environment. Interestingly, after 30 freeze–thaw cycles, many pores appear, and these pores are used to achieve positioning observation through SEM technology. Taking the S_2_ sample as an example, the relatively large pore B is used as the observation benchmark to achieve positioning observation. It can be seen from Figure 19 that the morphology of the S_2_ sample is significantly changed under freeze–thaw cycles. With the increase in freeze–thaw cycles, the pore area and roughness of the sample gradually increase. After 120 cycles, there is a significant honeycomb and pitted surface phenomenon.

Image processing analysis methods are used to analyze the morphology evolution of anhydrite rock under the freeze–thaw environment further. The image processing processes are as follows: (1) Grayscale image conversion: Converting the original image into an unsigned 8-bit grayscale image with pixels between 0 (black) and 255 (white). (2) Image denoising: Using the median filtering method to denoise the image. Median filtering is the most commonly used filtering method, which is a neighborhood operation that arranges the grayscale values of the image from small to large and uses the median as the output value of pixels to achieve noise reduction in the image, and the filtered image has clear edges. (3) Image binarization: The filtered image is converted into a binary image with pixels of 0 (black)—1 (white). (4) Edge detection: The primary purpose of edge detection is to obtain the range boundaries of pores and cracks in the damaged area. In this paper, the Sobel operator is used for edge detection, and the Sobel operator has good detection results for grayscale images with high noise.

The image processing results of pore B are shown in Figure 20. It can be seen clearly that the area and roughness of pore B significantly increase under the freeze–thaw cycle. To quantitatively determine the increase in pore area, the ratio of pore area to total image area in the binary image is defined as the proportion of pore area, and the calculated results are shown in Table 6. It can be seen that as the number of freeze–thaw cycles increases, the proportion of pore area in pore B gradually increases. After 120 cycles, the area proportion of pore B increases by 37.47% compared with 30 cycles.

## 4. Multi-Scale Deterioration Mechanism of Anhydrite Rock under Freeze–Thaw Environment

### 4.1. Correlation Analysis between Pore Structure and Macroscopic Mechanical Parameters

Porosity is a comprehensive parameter that reflects the pore structure of porous media. However, different pore structures influence the macroscopic mechanical parameter evolution of anhydrite rocks differently.

The relationship between pore area and mechanical strength is depicted in Figure 21 and Figure 22. As can be seen with the increase in the area of micropores, the mechanical strength of anhydrite rock increases exponentially. Meanwhile, with the increase in the area of mesopores and macropores, the mechanical strength decreases exponentially.

The mean correlation coefficient between the compression strength and the area of micropores, mesopores, and macropores is 0.75, 0.74, and 0.97, respectively. The correlation coefficient between the tension strength and the area of micropores, mesopores, and macropores is 0.75, 0.76, and 0.99, respectively. Thus, it can be concluded that macropores play the most significant role in the evolution of the mechanical parameters of anhydrite rock under the freeze–thaw environment.

### 4.2. Deterioration Mechanism Analysis

The mechanical loss rate is used to evaluate the deterioration of mechanical parameters of rocks that experience different freeze–thaw cycles. The mechanical loss rate can be calculated by Equation (4):(4)$$\mathrm{MLR}=\frac{\sigma_{0}-\sigma_{N}}{\sigma_{0}}\times 100\%$$
where MLR is the mechanical loss rate, $\sigma_{0}$ is the mechanical parameters of the untreated samples, and $\sigma_{N}$ is the mechanical parameters of samples treated with *N* freeze–thaw cycles.

Combined with the previous literature, it is found that the mechanical loss rate of anhydrite rock is higher than that of rocks with a similar porosity under the same freeze–thaw conditions. For example, when the number of freeze–thaw cycles is 60, the UCS loss rate of anhydrite rock and medium-grained feldspathic sandstone [42] is 27.23% and 6.6%, and the elastic modulus of anhydrite rock and sandstone [24] is 38.65% and 6.76%, respectively. When the number of freeze–thaw cycles is 120, the UCS loss rate of anhydrite rock and granite [43] is 46.54% and 13.9%, and the cohesion loss rate of anhydrite rock and sandstone [44] is 52.48% and 36.02%, respectively.

The above analysis indicates that when the initial porosity is similar, the damage to anhydrite rock under freeze–thaw cycles is more severe than that of other rocks. The freeze–thaw damage mechanism of rocks includes the volume expansion mechanism, hydrostatic pressure mechanism, capillary mechanism, and crystallization pressure mechanism. Zhou et al. [45] point out that the damage caused by capillary and crystallization pressure dominates when the rock freezing rate is low. On the contrary, when the freezing rate is high, the volume expansion and hydrostatic pressure mechanisms dominate. The damage mechanism of anhydrite rock is different from other rocks due to the complex water–rock interaction effect. It is believed that in addition to the freeze–thaw deterioration mechanisms mentioned above, water–rock expansion and water–rock dissolution deterioration effects also exist in the damage process of freeze–thawed anhydrite rock. The water–rock expansion effect is due to the volume expansion caused by the formation of dihydrate gypsum (CaSO_4_·2H_2_O) when anhydrite (CaSO_4_) meets water. The water dissolution effect is caused by the dihydrate gypsum (CaSO_4_·2H_2_O) dissolved in water. The connection between the mineral crystals is hydrolyzed, and the cohesion is reduced. The water–rock expansion and water–rock dissolution reaction equations are as follows:(5)$${\mathrm{CaSO}}_{4}+2H_{2}O\text{→}{\mathrm{CaSO}}_{4}\cdot 2H_{2}O$$
(6)$${\mathrm{CaSO}}_{4}\cdot 2H_{2}O\text{⇌}{\mathrm{Ca}}^{2+}+{\mathrm{SO}}_{4}^{2-}+2H_{2}O$$
(7)$${\mathrm{CaSO}}_{4}\text{⇌}{\mathrm{Ca}}^{2+}+{\mathrm{SO}}_{4}^{2-}$$

During the freeze–thaw process, different deterioration mechanisms promote each other, leading to the deterioration of the physical and mechanical properties of anhydrite rock. Therefore, when the porosity and saturation are similar, the freeze–thaw damage to anhydrite rock is more severe than other rocks, leading to a higher proportion of macropore structures under freeze–thaw action.

## 5. Conclusions

In this work, comprehensive multi-scale laboratory technologies involving mass variation, uniaxial compression, triaxial compression, Brazilian splitting, NMR, and SEM tests were performed to study the deterioration mechanisms of anhydrite rock under a freeze–thaw environment. As a result, the mass variation, main mechanical characteristics, porosity, and microstructure evolution of anhydrite were obtained. Then, the relationships between the microstructure and the mechanical characteristics were established, and the freeze–thawed deterioration mechanisms of anhydrite rock were revealed. The main conclusions can be summarized as follows:

(1)The macroscopic test results indicate that the mass variation increases exponentially with the increase in freeze–thaw cycles, while the mechanical strength, elastic modulus, and cohesion decrease exponentially. Meanwhile, as the freeze–thaw cycles increase, the frost resistance coefficient decreases, while the damage variable increases.(2)The porosity of anhydrite rock increases with the freeze–thaw cycles, and the mean porosity increases by 66.27% after 120 cycles. With the increase in freeze–thaw cycles, the area of micropores (r ≤ 0.1 μm) and PT-Ipore throat (0–0.1 μm) decreases exponentially. In comparison, the area of mesopores (0.1 μm < r < 1 μm), macropores (r ≥ 1 μm), and PT-II pore throat (0.1–4 μm) increases exponentially. Under the freeze–thaw treatment, the roughness of the sample gradually increases, and for the samples treated with 120 cycles, there is a significant honeycomb and pitted surface phenomenon.(3)The correlation analysis between microstructure and macroscopic mechanical parameters shows that macropores play the most significant role in the mechanical parameters evolution of anhydrite rock under the freeze–thaw environment.(4)It is found that the mechanical loss rate of anhydrite rock is higher than that of rocks with a similar porosity under the same freeze–thaw conditions. Finally, it is revealed that the water–rock expansion and water dissolution effects play a crucial role in the multi-scale damage of anhydrite rock under a freeze–thaw environment.

## Figures and Tables

**Figure 1 materials-17-00726-f001:**
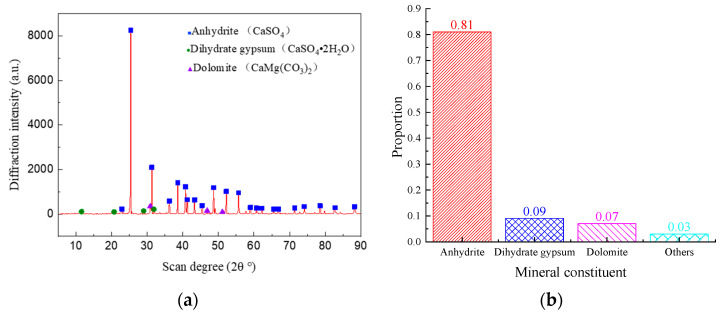
Mineral composition of anhydrite rock: (**a**) X-ray diffraction pattern and (**b**) proportion of mineral mass.

**Figure 2 materials-17-00726-f002:**
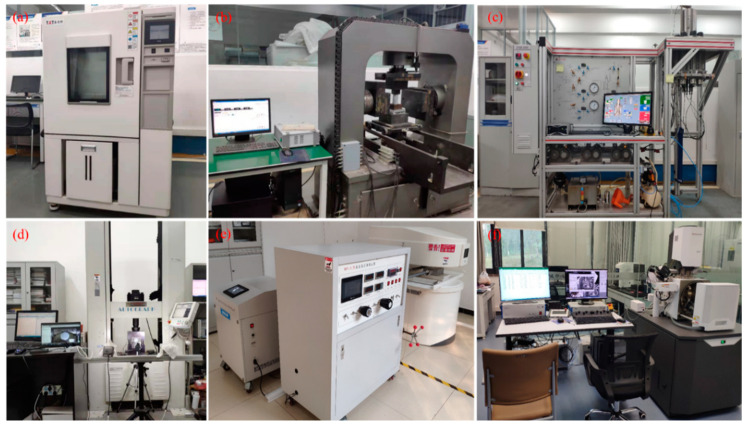
Main instruments: (**a**) high- and low-temperature alternating damp heat test chamber, (**b**) WDAJ-600 microcomputer-controlled electro-hydraulic servo rock shear rheological test machine, (**c**) ROCK600-50HTPLUS multifunctional rock full stress triaxial test system, (**d**) AG-250kN IS precision electronic universal material test machine (**e**) non-destructive NMR system for rock core imaging and analysis, and (**f**) field emission transmission electron microscope.

**Figure 3 materials-17-00726-f003:**
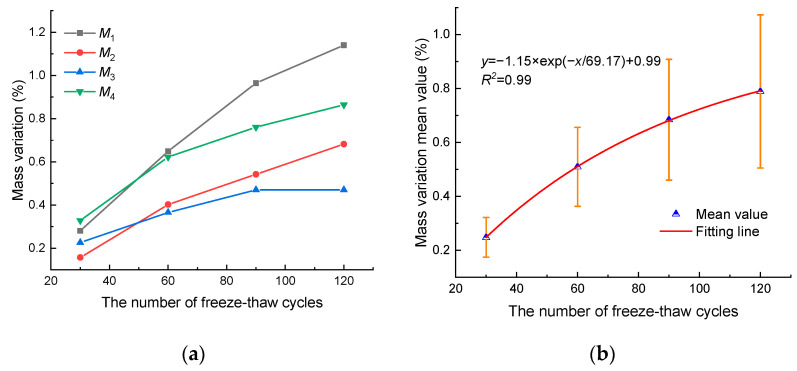
Mass variation evolution: (**a**) mass variation of four samples and (**b**) mass variation mean value.

**Figure 4 materials-17-00726-f004:**
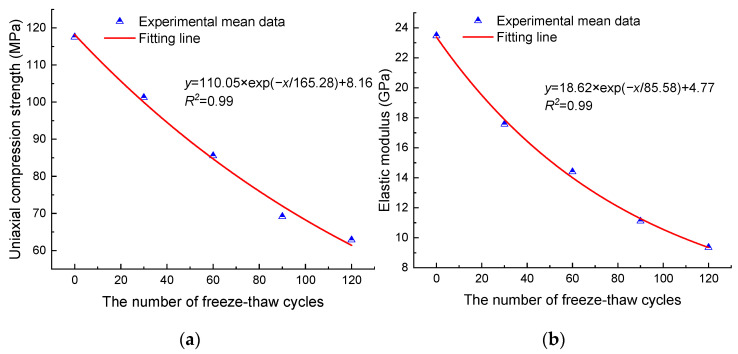
Uniaxial compression mechanical properties: (**a**) uniaxial compression strength and (**b**) elastic modulus.

**Figure 5 materials-17-00726-f005:**
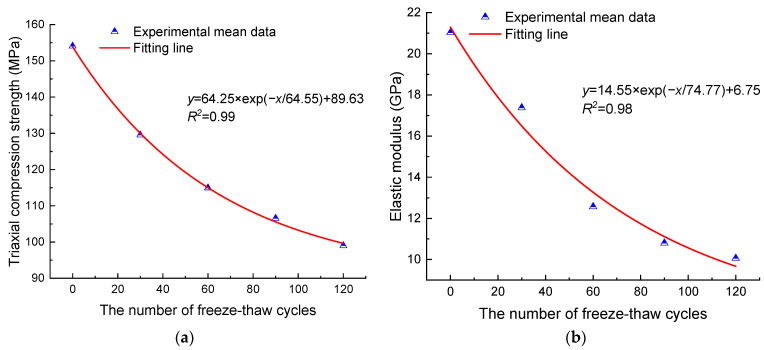
Triaxial compression mechanical properties with a confining pressure of 3 MPa: (**a**) triaxial compression strength and (**b**) elastic modulus.

**Figure 6 materials-17-00726-f006:**
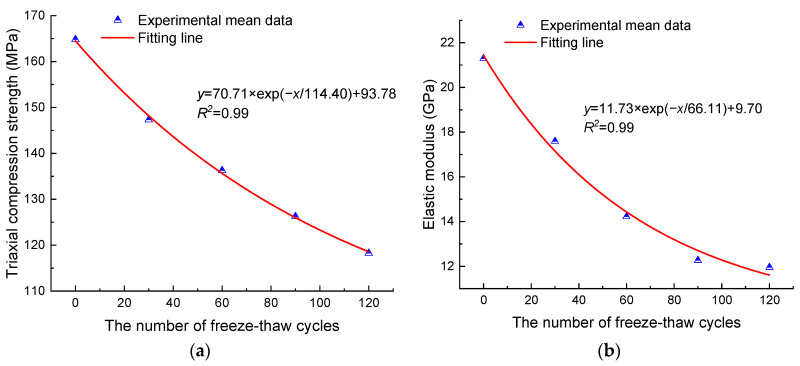
Triaxial compression mechanical properties with a confining pressure of 5 MPa: (**a**) triaxial compression strength and (**b**) elastic modulus.

**Figure 7 materials-17-00726-f007:**
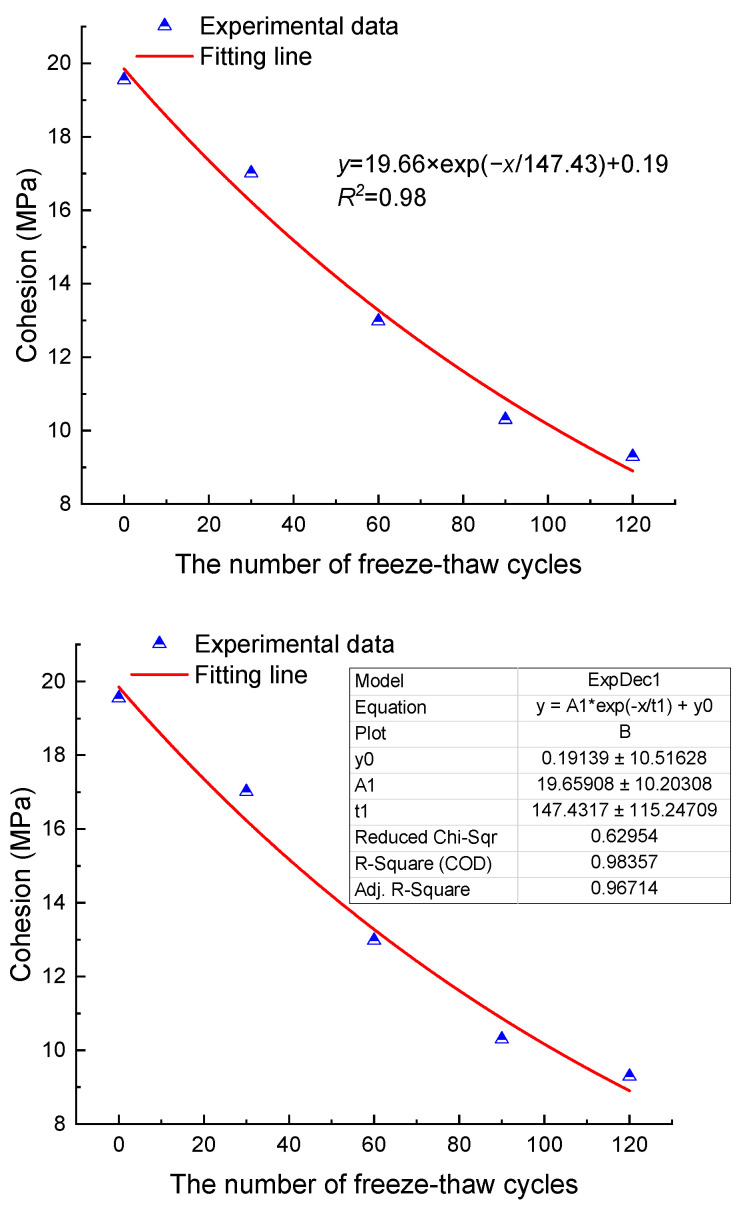
Relationship between the cohesion and the number of freeze–thaw cycles.

**Figure 8 materials-17-00726-f008:**
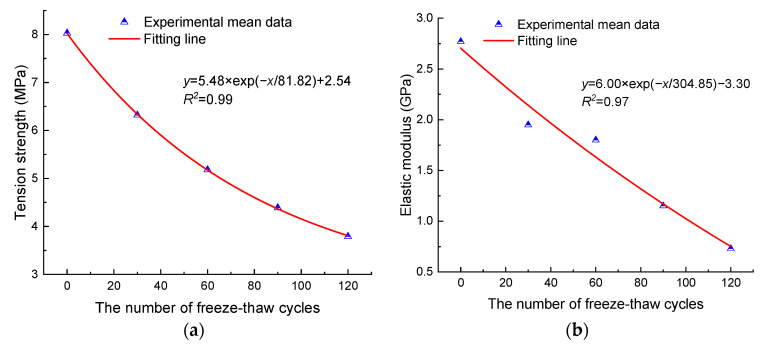
Tension mechanical properties (**a**) tension strength and (**b**) elastic modulus.

**Figure 9 materials-17-00726-f009:**
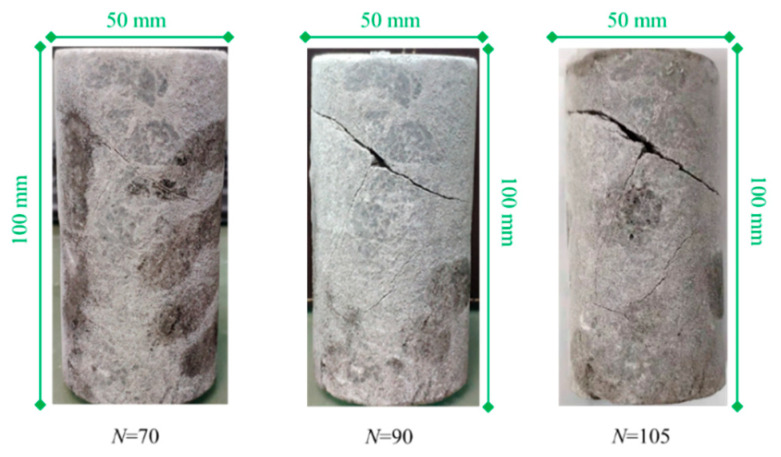
Macro damage phenomenon of the sample during freeze–thaw cycles.

**Figure 10 materials-17-00726-f010:**
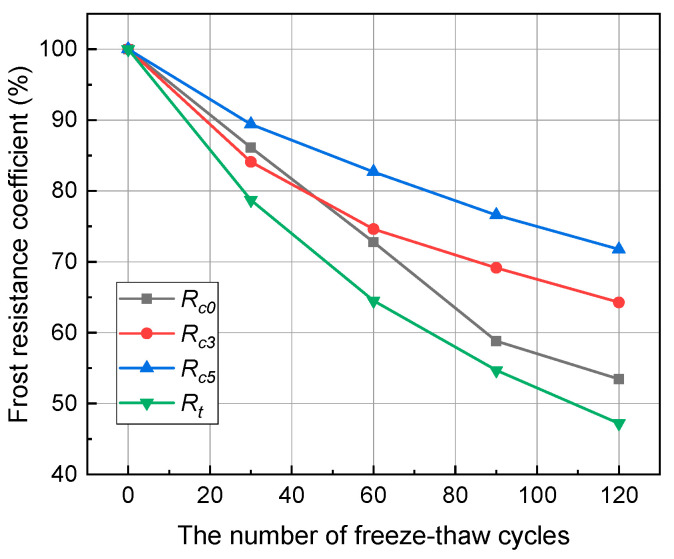
Relationship curves between frost resistance coefficient and freeze–thaw cycles.

**Figure 11 materials-17-00726-f011:**
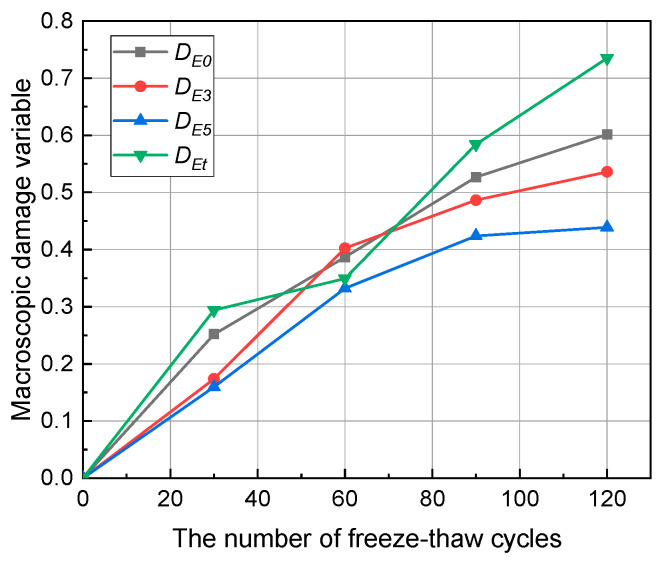
Macroscopic damage variable evolution of anhydrite rock under freeze–thaw cycles.

**Figure 12 materials-17-00726-f012:**
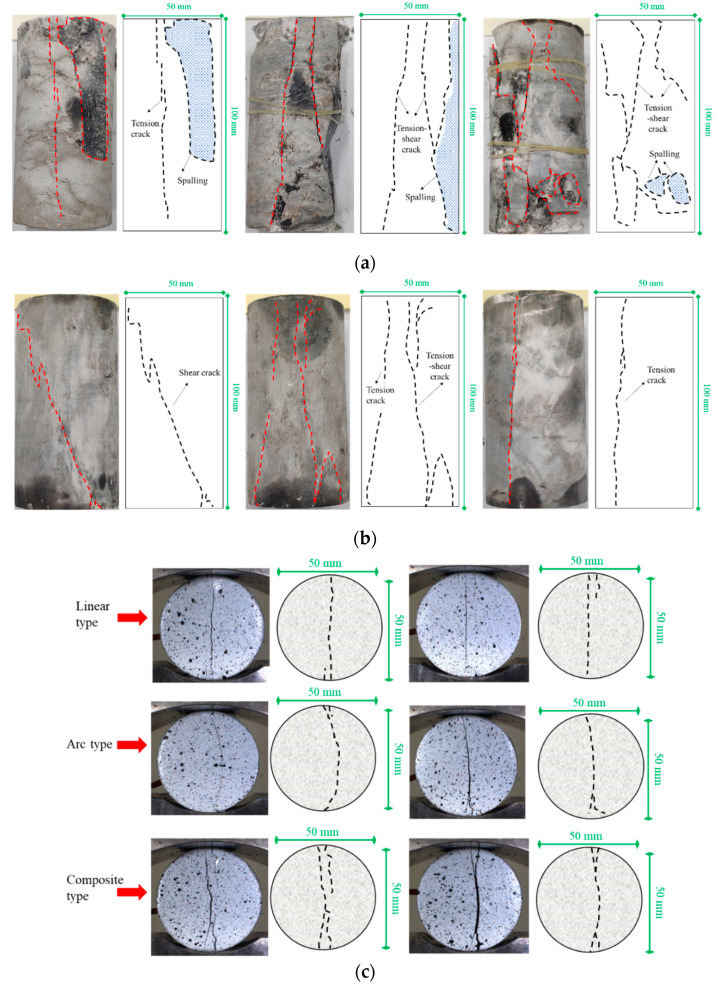
Cracking modes of anhydrite rock under different loading conditions: (**a**) uniaxial compression, (**b**) triaxial compression, and (**c**) Brazilian splitting.

**Figure 13 materials-17-00726-f013:**
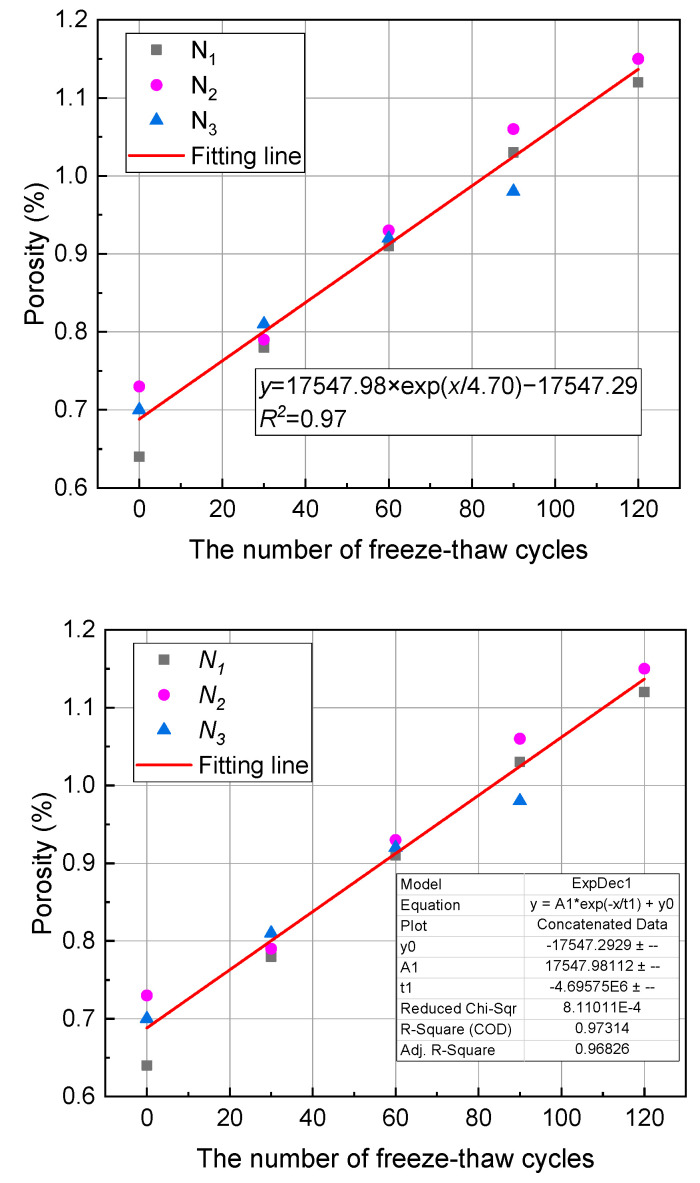
Porosity evolution of anhydrite samples under freeze–thaw cycles.

**Figure 14 materials-17-00726-f014:**
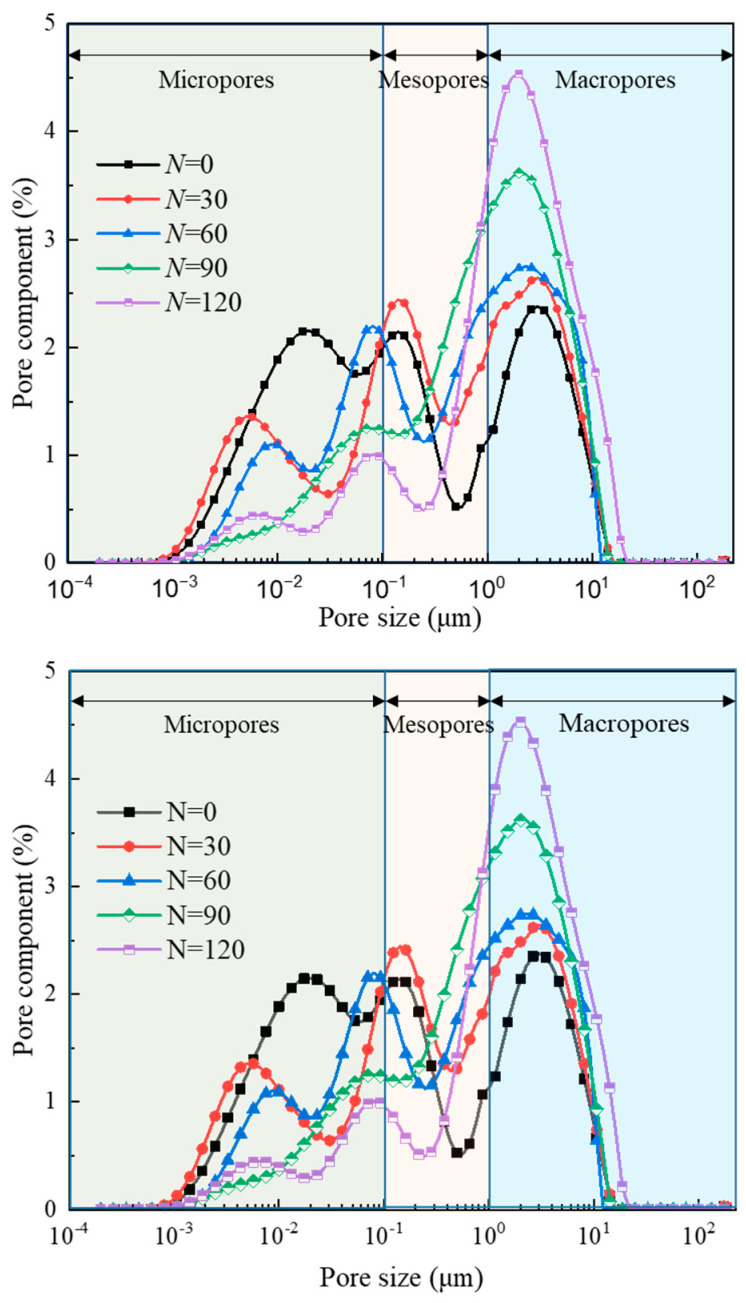
Pore size distribution evolution curves of anhydrite rock under freeze–thaw cycles.

**Figure 15 materials-17-00726-f015:**
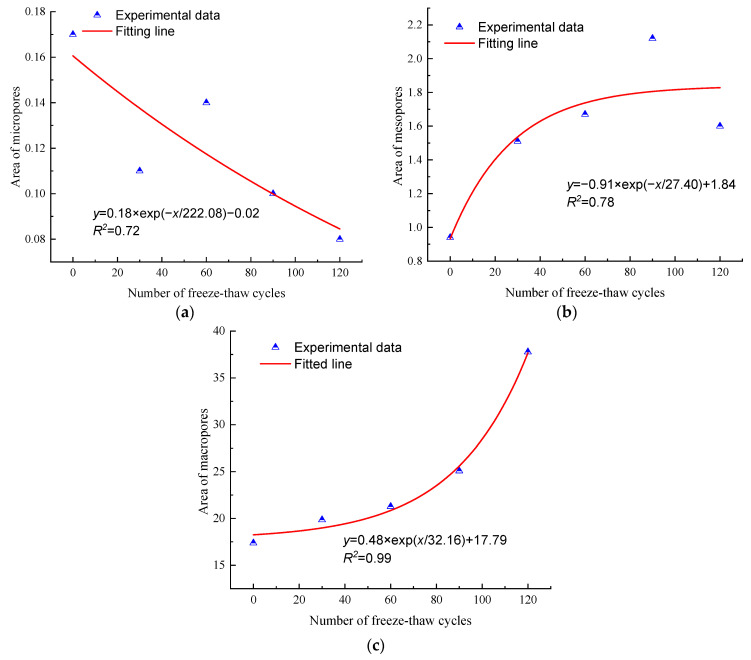
Curve area evolution law for different types of pores of N_1_ sample under freeze–thaw cycles: (**a**) micropores, (**b**) mesopores, and (**c**) macropores.

**Figure 16 materials-17-00726-f016:**
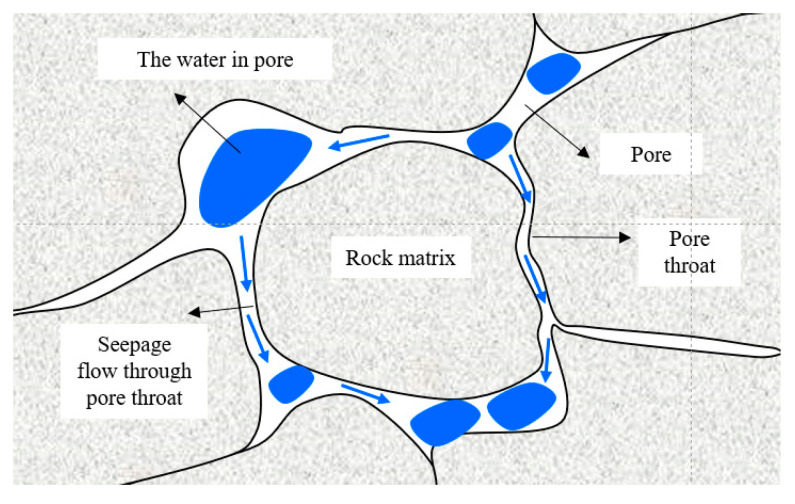
The schematic diagram for the pore structures of rock.

**Figure 17 materials-17-00726-f017:**
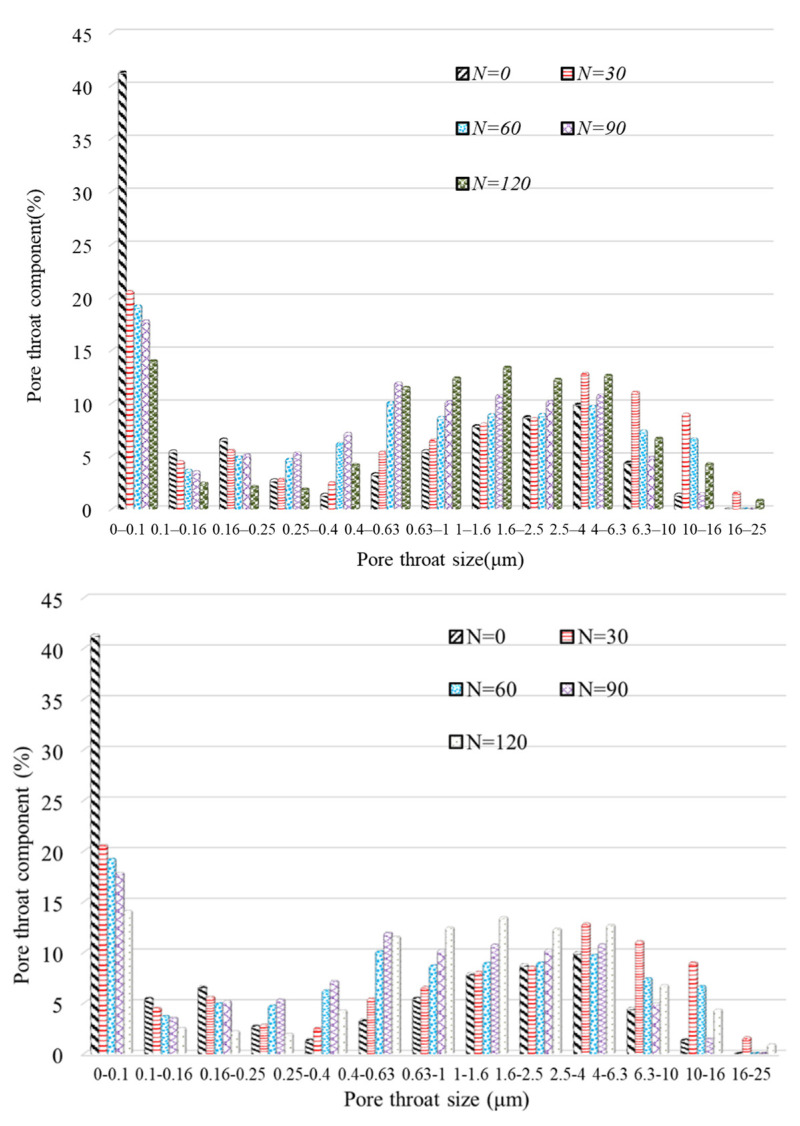
Pore throat distribution evolution of anhydrite rock under freeze–thaw cycles.

**Figure 18 materials-17-00726-f018:**
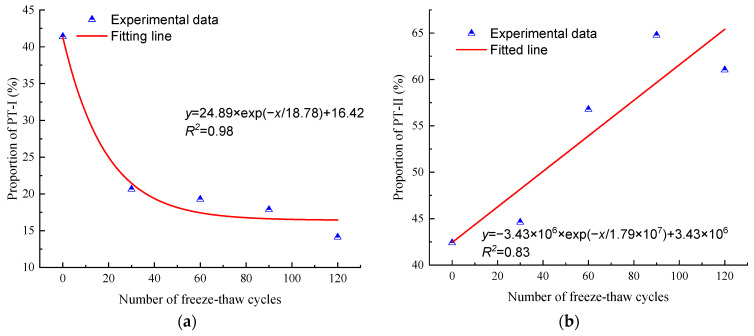
Evolution of different pore throats in anhydrite rock under freeze–thaw cycles: (**a**) PT-I and (**b**) PT-II.

**Figure 19 materials-17-00726-f019:**
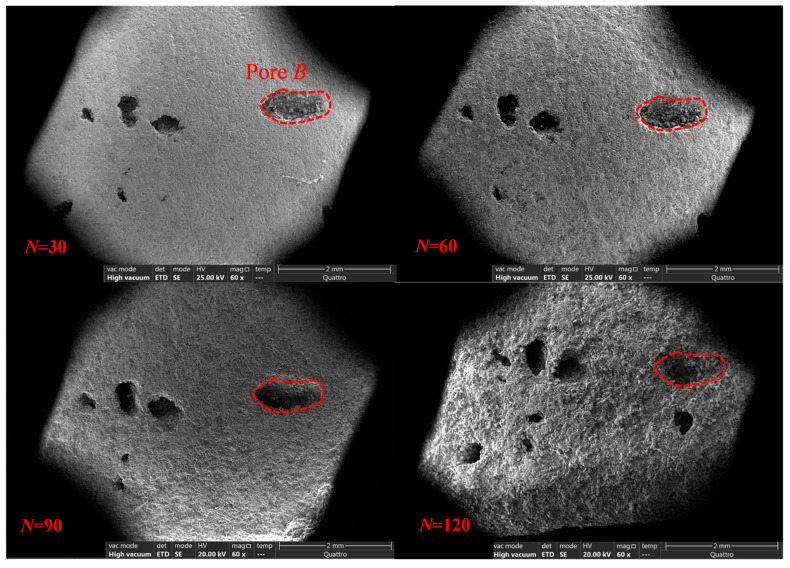
Morphology evolution of anhydrite sample under freeze–thaw cycles.

**Figure 20 materials-17-00726-f020:**
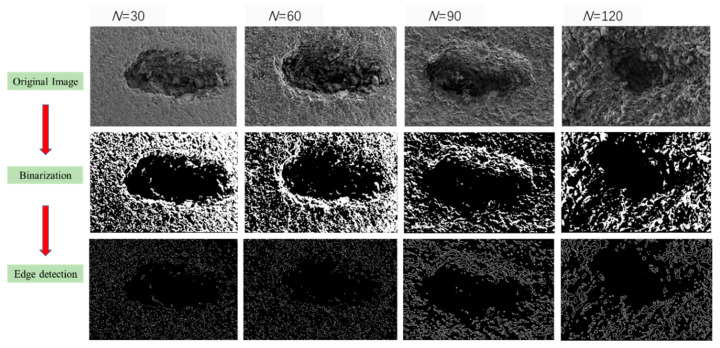
Image processing analysis of pore B under freeze–thaw cycles.

**Figure 21 materials-17-00726-f021:**
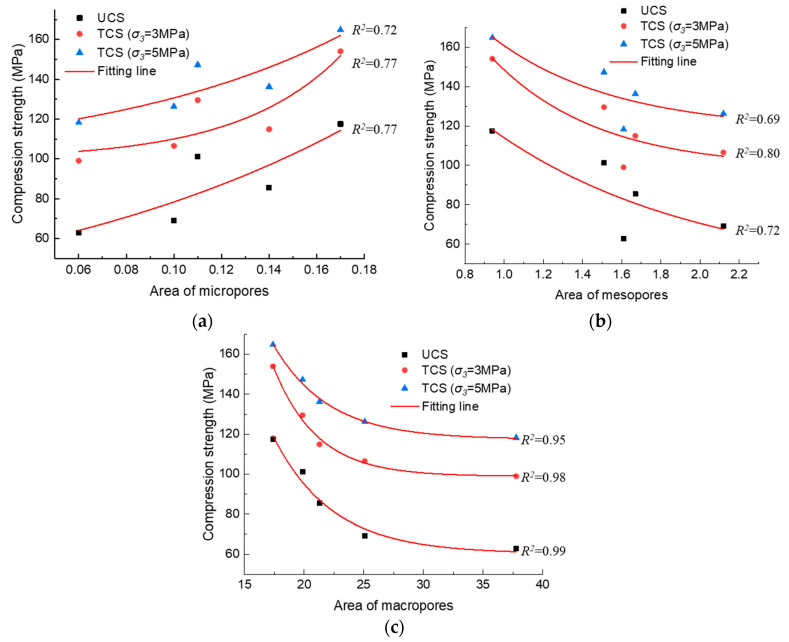
Relationship between pore area and compression strength: (**a**) area of micropores, (**b**) area of mesopores, and (**c**) area of macropores.

**Figure 22 materials-17-00726-f022:**
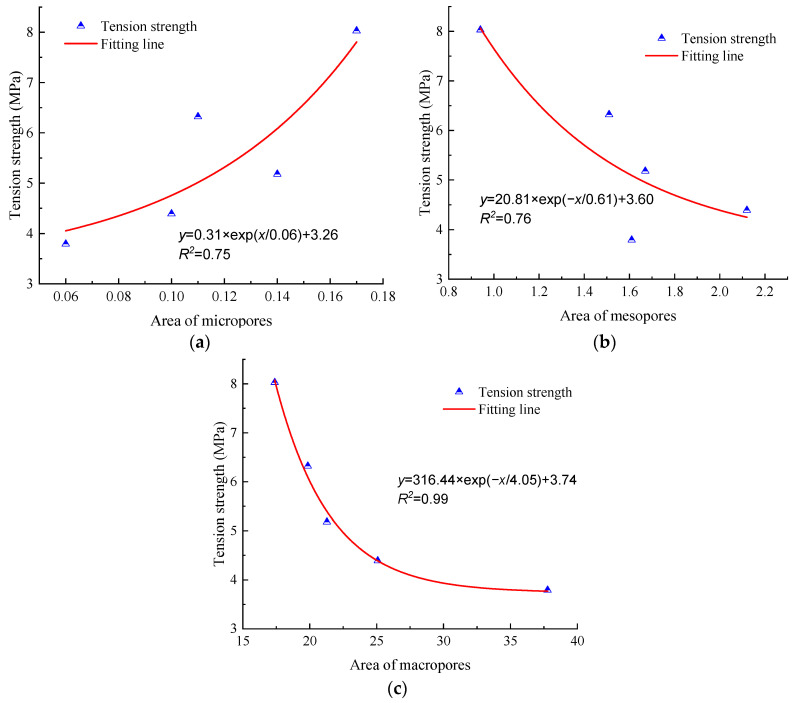
Relationship between pore area and tension strength: (**a**) area of micropores, (**b**) area of mesopores, and (**c**) area of macropores.

**Table 1 materials-17-00726-t001:** Results of uniaxial compression tests.

Sample Number	Uniaxial Compression Strength (MPa)					Elastic Modulus (GPa)				
	0	30	60	90	120	0	30	60	90	120
1	113.68	104.96	85.21	69.23	61.80	23.58	18.30	13.54	11.77	9.07
2	121.67	99.72	82.99	67.30	59.99	23.27	16.06	13.82	9.98	9.55
3	117.28	98.99	88.41	70.88	66.74	23.57	18.34	15.83	11.58	9.43
Mean value	117.54	101.22	85.53	69.14	62.84	23.47	17.57	14.40	11.11	9.35

**Table 2 materials-17-00726-t002:** Results of triaxial compression tests ($\sigma_{3}=3\;\mathrm{MPa}$).

Sample Number	Triaxial Compression Strength (MPa)					Elastic Modulus (GPa)				
	0	30	60	90	120	0	30	60	90	120
1	151.33	133.84	119.88	103.63	97.17	21.77	18.69	14.49	11.53	10.04
2	156.76	125.22	109.99	109.44	100.86	20.28	16.06	10.64	10.06	10.08
Mean value	154.05	129.53	114.94	106.54	99.02	21.03	17.38	12.57	10.80	10.06

**Table 3 materials-17-00726-t003:** Results of triaxial compression tests ($\sigma_{3}=5\;\mathrm{MPa}$).

Sample Number	Triaxial Compression Strength (MPa)					Elastic Modulus (GPa)				
	0	30	60	90	120	0	30	60	90	120
1	168.02	144.91	139.49	125.37	115.50	22.38	18.50	13.82	12.58	11.84
2	161.55	149.71	133.03	127.11	121.02	20.20	16.68	14.61	11.95	12.05
Mean value	164.79	147.31	136.26	126.24	118.26	21.29	17.59	14.22	12.27	11.95

**Table 4 materials-17-00726-t004:** Cohesion and internal friction angle of anhydrite rock under different freeze–thaw cycles.

Freeze–Thaw Cycles					
	0	30	60	90	120
*c* (MPa)	19.55	17.01	12.98	10.3	9.29
*φ* (°)	54.03	53.11	55.61	57.15	56.98

**Table 5 materials-17-00726-t005:** Brazilian splitting test results of anhydrite rock under different freeze–thaw cycles.

Sample Number	Tension Strength (MPa)					Elastic Modulus (GPa)				
	0	30	60	90	120	0	30	60	90	120
1	8.57	6.02	5.02	4.20	3.86	2.92	1.78	1.70	1.05	0.71
2	7.91	6.26	5.48	4.57	3.43	2.65	2.00	1.80	1.27	0.64
3	7.61	6.69	5.04	4.41	4.08	2.73	2.08	1.90	1.13	0.85
Mean value	8.03	6.32	5.18	4.39	3.79	2.77	1.95	1.80	1.15	0.73

**Table 6 materials-17-00726-t006:** Proportion of pore area for anhydrite rock under freeze–thaw cycles.

Number	Proportion of Pore Area (%)			
	30	60	90	120
*B*	59.46	66.38	78.91	81.74

## Data Availability

The data presented in this study are available on request from the corresponding author.

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
