# Peer review of "Investigation of the Multi-Scale Deterioration Mechanisms of Anhydrite Rock Exposed to Freeze–Thaw Environment"

_materials, 2024, doi:10.3390/ma17030726_

Round 1

Reviewer 1 Report

Comments and Suggestions for Authors

1. According to Figure 3 (a), only sample M2 shows a different trend in the relationship between the number of freeze-thaw cycles and mass variation. What could be the reason for this? Could there be structural defects unique to sample M2 compared to other samples?

2. Overall, in the graphs with a fitting line, it seems that a high R2 value was obtained by performing fitting using the mean value. How would the R2 value change if individual values were used instead of the mean value?

3. During the triaxial compression test, the maximum confining stress value appears to be somewhat low. Is there a reason for setting it this low?

4. During the triaxial compression test, it is anticipated that the duration of the test increases with the magnitude of the confining stress. What is the temperature during the test, and could this temperature potentially affect the strength measurements?

5. To more clearly analyze the relationship between the internal friction angle and the number of freeze-thaw cycles, it is judged that more tests should be conducted to increase the test set.

Comments on the Quality of English Language

The quality of English expression needs improvement to ensure that readers comprehend the research objectives and results.

Reviewer 2 Report

Comments and Suggestions for Authors

This manuscript provides an experimental research on the weathering effects of freezing-thawing cycles on anhydrite rock. The experiment is well designed, the text is mostly clear and results reliable. The subject is also suitable for the journal and I recommend the publication of the manuscript but I think that some previous improvements are needed to get a better understanding of the study. I include some suggestions to improve it.

INTRODUCTION

In the second line of the first paragraph of the introduction, the authors write “Rocks in nature consist of mineral grains, cementing substances, …”. I do not understand what they mean with “rocks in nature”. All rocks are in nature. Moreover, rocks consist of mineral grains, but not all have cementing substances: most igneous and metamorphic rocks have not cementing substances… This definition can be valid for detrital sedimentary rocks, but not the rest of rocks.

In the second paragraph of the introduction the authors provide some references of studies of freezing-thawing effects on stones. They provide recent references, but there are much others that are classical studies. Indeed, the effect of freezing-thawing cycles have been studied for a long time… The authors refer to the study of Deprez et al (2020) and there are references there in, and other references that can be useful to complete the introduction. For instance, J. Martínez-Martínez, D. Benavente, M. Gomez-Heras, L. Marco-Castaño, M.Á. García-del-Cura. Non-linear decay of building stones during freeze–thaw weathering processes. Constr. Build. Mater., 38 (2013), pp. 443-454, 10.1016/j.conbuildmat.2012.07.059

I strongly recommend to provide a short list of stone types that has been studied. The purpose is to highlight the value of this manuscript as, as far as I know, anhydrite rock has not been studied!

In the third paragraph the authors provide several techniques. I recommend remove the techniques that are not used in this manuscript.

MATERIALS AND METHODS

In section 2.2.2. please, provide more data about the mechanical resistance tests.

EXPERIMENTAL RESULTS AND ANALYSIS

I have some objections to figure 3b. This was ploted with the average values of figure 3A. However, M2 provide some anomalous data that the authors comment. I think this data should be removed to assess the average. Moreover, the plot should provide the error bars. This is very important. With the figure 3b in the present format, the results suggest that the mass variation is linear with the number of cycles, and this is not true, the typical curve is exponential. Please, change this and clarify.

MULTI-SCALE DETERIORATION MECHANISM OF ANYDRITE ROCK UNDER FREEZE-THAW ENVIRONMENT

In section 4.2. The authors refer to a sandstone. What do they mean? Have they tested the same experiment with a sandstone. I think that probably the try to compare results of anhydrite with sandstone, but they show a SEM figure (Fig. 23) with results in anhydrite and sandstone. Please, if you have used a sandstone in experiments, explain this, and provide data on the composition and porosity of the sandstone. Sandstone is a very different rock, and it has been previously study the effect of freezing-thawing cycles, so such data from references should be used for comparison.

Round 2

Reviewer 2 Report

Comments and Suggestions for Authors

The authors have addressed all the suggestions and comments. I find that the manuscript is ready for publication.